# Effects of Nano Aluminum Powder on the Mechanical Sensitivity of RDX-Based Explosives

**DOI:** 10.3390/nano11092182

**Published:** 2021-08-25

**Authors:** Jun Dong, Weili Wang, Xiaofeng Wang, Qiang Zhou, Run Miao, Maohua Du, Bo Tan, Yuanjing Wang, Tengyue Zhang, Yafei Li, Fangjie Cao

**Affiliations:** 1Department of Engineering, Naval University of Engineering, Wuhan 430033, China; welcome204dj@163.com (J.D.); miaorun1769@163.com (R.M.); 18207157778@139.com (M.D.); Turbo1996@126.com (B.T.); 2Xi’an Modern Chemistry Research Institute, Xi’an 710065, China; 13319289851@sina.com; 3China Academy of Ordnance Science, Beijing 100089, China; zqpcgm@gmail.com (Q.Z.); wyjlucky2021@163.com (Y.W.); zhangtengyue0201@163.com (T.Z.); 4Navy Research Institute, Beijing 100072, China; asiafly@163.com

**Keywords:** aluminized explosive, nano-Al, micron-Al, mechanical sensitivity, scanning electron microscopy

## Abstract

As nano-aluminum powder (NAP) can improve the detonation performance of aluminum-containing explosives, more and more explosives with NAP as the metal ingredient have been studied. It is believed that the mechanical sensitivity of explosives can be significantly enhanced by the added nano-sized aluminum powder. However, the mechanism for the enhancement has not been clarified. In order to illuminate the effects of NAP on the mechanical sensitivity of explosives, two RDX-based aluminum-containing explosives with the same weight ratio and preparation process were investigated despite the aluminum powders with different nano-size and micron-size. The morphology and surface atomic ratio of the two explosives were examined by scanning electron microscopy with energy dispersive spectroscopy tests. The contact angle and other microstructures properties of the explosives were calculated by Material Studio software. Results revealed that the impact and friction activity was determined by the aluminum particle sizes and explosive components. This paper clarified the mechanism for the increase in explosives sensitivity by the addition of NAP, which provide reference for the scientific and technical design of novel explosives.

## 1. Introduction

For aluminized explosives, aluminum powder can release a large amount of reaction heat in the secondary reaction following the detonation wave, which increases the detonation heat and specific volume of the explosive during detonation, and also enable the explosive to have a higher work capacity [1,2,3,4,5,6,7,8,9,10,11,12,13]. That is why it is called high-power explosive or high explosive and has become one of the main research categories in the explosives field. In the study of the mechanism of the detonation reaction between aluminum powders and explosives, the secondary reaction theory of aluminum-containing explosives is proposed: the Al powder in the formulation does not participate in the reaction on the wavefront during the detonation [14,15,16,17,18,19,20,21,22]. For the energy release reaction on the detonation wave front, Al powder is an inert endothermic substance, which will absorb the released energy by the explosive reaction from the detonation wave front. At the same time, the powder has good thermal conductivity (the thermal conductivity coefficient is 230 W/(m·K^−1^)). With the rapid expansion of the detonation products, part of the energy on the detonation wavefront is quickly transferred to the air. This will make the energy that supports the detonation propagation of explosives decrease, which is macroscopically reflected in the decrease of detonation pressure and detonation velocity of the aluminized explosives with the increase in Al powder content [23]. In order to improve the detonation energy output structure of the aluminized explosives, the time for the Al powder to participate in the detonation reaction is advanced.

Chen Lang [24] found that the size of aluminum powder had a significant effect on the detonation performance of explosives. Small-sized aluminum powder is more likely to react with the explosive detonation products with an advanced reaction time, large reaction volume, and fast energy release, which enhances the work capacity of explosives. Especially the addition of nano-sized aluminum powder can effectively improve the work capacity of explosives. Leonid Kaled et al. [25] studied Alex nano-aluminum particles and their applications in liquid and solid rocket propellants. It is found that the most effective way to achieve complete combustion is the application of aluminum powder with a particle size at least 1 to 2 orders of magnitude smaller than the one used in conventional solid propellants. Therefore, research on the application of nano-aluminum powder in explosives has attracted more and more attention in recent years. Some researchers have systematically studied the characteristics of RDX-based explosives, such as detonation heat [26], detonation velocity [27], metal acceleration ability [28], underwater explosion energy [29], front curvature effect [30], etc. Some papers also reported the influence of nano-Al on the sensitivities of cyclo-trimethylene trinitramine (RDX)-based explosives [31], which clearly showed that the addition of nano-Al increased the mechanical sensitivity and flame sensitivity of the explosive. However, the research in this field is not deep enough. The results just reflected the macroscopic performance of explosives and some basic theories. The role of microstructure on the materials’ performance and relevant scientific theory are yet to be clarified. Until now, the mechanism of NAP inducing the increase in the mechanical sensitivity and flame sensitivity of the explosives has not been understood. Previous results have not given enough illustration on the improvement and design of nano-Al-containing explosive formulations.

Reducing the mechanical sensitivity of explosives is still a very effective way to improve the intrinsic safety of explosives manufacturing, transportation, storage and other production links. Research on the influencing factors of the explosives’ mechanical sensitivity can provide guidance for the improvement of the explosive performance from the aspects of explosive formulation design, material pretreatment, the manufacturing process, etc.

To further reveal the influence of NAP on the explosives’ mechanical sensitivity, the RDX-based explosives with the same component ratio containing nano-sized and micron-sized aluminum powder were prepared with the same process, separately. The surface atomic energy spectra, contact angle and other properties were examined. Material Studio (MS) software was utilized for modeling the structure and calculating the structure parameters. The correlation between the structure-activity, aluminum powders size, explosive components and the impact-, friction-sensitivity were clarified. The results of the paper provided scientific reference for the safety application of nano-aluminum powder in explosives in the future.

## 2. Experiment and Calculation

### 2.1. Reagents and Instruments

Nano-aluminum powder (purity 99.8%) with the average particle size of 100 nm made by the Xi’an Modern Chemistry Research Institute. Micron-aluminum powder (purity 98.9%) with an average particle size of 20 μm made by Xi’an Aerospace Chemical Propulsion Co., Ltd. RDX with an average particle size of 123.6 μm was produced by Gan-su Yin-guang Chemical Industry Group Co., Ltd. Paraffin wax with a melting point of 68 °C was produced by Petro China Fu-shun Petrochemical Company.

The friction and impact sensitivity instrument (H3.5–10W) was employed to examine the sensitivity of explosives. A FEI QUANTA 600 field emission scanning electron microscope (FE-SEM) that was manufactured by FEI Co. USA was employed to investigate the explosive morphology. The accelerating voltage was 10 kV and the tests were conducted in high vacuum mode. N_2_ adsorption–desorption measurements were carried out at 77 K using a Quantachrome Autosorb gas-sorption system to determine the specific surface area of nano-Al, micro-Al and RDX, separately. The specific surface area of nano-Al, micron-Al and RDX are 22.22, 0.55 and 0.344 m^2^/g. The transmission electron microscopy (TEM) tests of nano-Al was operated by using a Hitachi (H 9000 NAR).

### 2.2. Explosive Preparation

RDX-based explosives containing micron-aluminum and nano-aluminum powders are designed, respectively. The morphology of nano-Al and micron-Al powder are shown in Figure 1. Figure 1A,B clearly shows that the nano-Al shows a deeper black color; thus, the micron-Al shows a grey color. Figure 1C showed the TEM test results of nano-Al powder, which clearly exhibited that the nano-Al were in ball-like structures. The diameter of the nano-Al are also examined and the results are showed in Figure 1D. It is obviously shown that the diameter of nano-Al powder is located in a large range despite the major diameter being about 150 nm.

Paraffin wax is used as the insensitive agent to reduce the sensitivity of RDX. The component and proportions of the explosives are shown in Table 1.

Explosive 1 and 2 were prepared by the same process: the paraffin wax was dissolved by petroleum ether, and then the RDX and aluminum powders were added, respectively. All the reagents were mixed in a kneader for 30 min. Then, the mixture was poured out and the solvent was volatilized until the explosive reached the semi-dry state, which means that there is no petroleum ether could be volatized anymore, but the state still looks a little wet. The obtained explosives were screened and granulated with an 8-mesh screen mesh. The finally obtained explosive molding powders were fully dried and adopted to examine the performance. The photo morphology of the two explosive are shown in Figure 2. Figure 2 showed that the explosive 1 with the nano-Al shows a deeper color compared with the explosive with micron-Al.

### 2.3. Performance Test

#### 2.3.1. Impact Sensitivity Test

The impact sensitivity of explosives is measured by the calibrated H3.5–10W drop-hammer impact sensitivity instrument. The weight of the drop hammer was 10 kg, and the weight of the explosive is 50 mg. The examination was conducted according to the drop height of each test. The 50% explored drop height of the tests is believed to be the impact sensitivity. The tests are divided into two groups, and each group has 25 rounds. Additionally, the final result is the even value of all the test rounds in each group.

#### 2.3.2. Friction Sensitivity Test

The friction sensitivity of the explosives is determined by the surface pressure of 3.92 MPa by the swing angle of 90° with the weight of 50 mg. The tests are also divided into two groups with 25 rounds in each group. The average value of each group is obtained.

### 2.4. Initial Structure Selection and Optimization Calculation of Explosive Components

#### 2.4.1. RDX

RDX is a molecular crystal. The crystal was sliced and a vacuum layer of 20 Å was directly added for calculation by using the Vienna Ab initio Simulation Package (VASP), combined with plane-wave basis sets. The adopted plane wave cut-off energy in the calculation is 520 eV. The electronic self-consistency and the convergence criteria of the force between atoms adopts the default accuracy of VASP. Meanwhile, the atoms in the lower half are kept in place, and those in the upper half are optimized. All operations on the Brillouin zone of the primitive cells use a 3 × 3 × 1 Monhkorst-Pack K-point grid centered on Γ. Van der Waals force is corrected by the DFT-D3 method. The exchange–correlation functional between electrons is calculated using the Perdew–Burke–Ernzerh (PBE) method in the generalized gradient approximation (GGA). The calculated energy is −1043.3066 eV.

#### 2.4.2. Al_2_O_3_

The calculation is started with α-Al_2_O_3_ (space group: R-3C) as the initial structure. The energy of the seven crystal planes (111), (110), (101), (011), (100), (010) and (001) that are cut with different thicknesses are calculated, respectively. During cutting, there are four layers of atoms in the lower layer, namely, a total of 40 atoms. The calculation is conducted using the VASP software package combined with the method of plane-wave basis sets. The plane-wave cut-off energy used in the calculation is 520 eV and the convergence standard for electronic self-consistency is 10^−6^ eV. The thickness of the vacuum layer in the vertical direction is 20 Å. All operations on the Brillouin zone of the primitive cells use a Monkorst-Pack K-point grid centered on Γ. The positions of the upper two layer atoms (the lower two layers of atoms are kept in place) were fully optimized so that the force between the atoms is less than 10^−3^ eV/Å. In this paper, the GGA-PBE is applied to calculate the exchange-correlation functional between electrons. The specific data of the calculated results are shown in Table 2.

Based on the calculation, it can be concluded that the lowest energy among the several planes appears in three crystal planes (110), (101) and (011), which is approximately −293.38 eV. Considering that the Al_2_O_3_ is a crystal with high symmetry, there is not much difference between (110), (101) and (011) crystal planes. Therefore, the optimized results by the sixth cutting method for the relatively common (110) crystal plane is selected as the reference structure during adsorption. So, paraffin is placed on (110) surface to calculate the adsorption energy.

#### 2.4.3. Paraffin Molecule

The initial structure of the paraffin molecule is manually constructed by the Material Studio software according to the schematic diagram given in the literature. Additionally, then, the structure is optimized using the VASP software package. Considering that the adsorption of paraffin molecules on the alumina (110) surface must be calculated, the cell must be expanded and combined with the actual situation of the computing resources. A paraffin molecule containing 10 carbon atoms is selected for the simulation. The calculation software and the selected pseudopotential are the same as those selected for paraffin and RDX. Because only a single adsorbed molecule is selected, the interaction between molecules is excluded when the molecule is optimized. The molecules are placed in a large cube cell (along the diagonal), and the cell is expanded to make sure that the molecular spacing is about 15–20 Å. In order to eliminate the interaction between adjacent molecules, cell calculation (ISIF = 2) is performed when the structure is optimized. For the Brillouin zone calculations of the structure, a 1 × 1 × 1 Monhkorst-Pack type K-point grid centered on Γ is adopted. In addition, the Van Der Waals force is corrected by the DFT-D3 method, and the exchange–correlation functional between electrons is calculated by the GGA-PBE method. Finally, it is drawn that the energy obtained after optimization is −173.074 eV.

#### 2.4.4. Al_2_O_3_ + Paraffin Molecule

In order to eliminate the periodic conditions induced interactions between paraffin molecules, cells for expansion are selected when paraffin molecules are adsorbed on the alumina surface. The initial and optimized structure of the adsorbed molecules after expansion are shown in Figure 3 (672 atoms). The top view of the structure shows that the molecular structure of paraffin wax is adsorbed on the surface of alumina. The original structure of Figure 3 shows the original structure, and the optimized structure of Figure 3 shows the optimized structure. In the calculation, the two layers on the surface of the alumina are optimized, and the two below layers are fixed. The paraffin molecules are completely optimized. In the initial structure, the vertical distance between the paraffin molecules and the alumina surface is about 0.5 Å. Following the optimization, a repulsive effect between the two structures compared to the initial one is produced. Under the condition, the distance between the two molecules is about 2.4 Å as shown in Figure 3. The energy of the optimized structure is −4950.088 eV.

## 3. Results and Discussion

### 3.1. Microscopic Morphology and Surface Elemental Ratio of Explosives

The morphology and energy dispersive spectrum (EDS) test results of the two explosives are shown in Figure 4/Table 3 and Figure 5/Table 4, separately. The elemental composition of the two explosives are normalized both in weight and atomic percentage.

Comparison between Table 3 and Table 4 showed that the component ratios of Al in the two explosives were different from each other despite the initial weight ratio of Al in the two explosives were same. This should be attributed to the different size distribution of the two explosives. The specific surface area of NAP in explosive 1 is 10^4^ times higher than that of micron Al in explosive 2. Though, in theory, the Al content in explosive 1 should be higher than that in explosive 2, the actual test results are quite different from the theoretical calculation results. This phenomenon should be originated from the serious agglomeration of nano-Al in explosive 1. The SEM test results of explosive 1 and 2 are showed in Figure 6 and Figure 7, respectively. Figure 6 obviously revealed that nano-Al in explosive 1 did not disperse uniformly.

SEM tests of micron-Al and explosive 2 are provided in Figure 7.

Comparison of Figure 6 and Figure 7 showed that the diameter of nano-Al was much smaller than that of micron-Al despite both of them showing perfect spherical morphology. On the other hand, it is also clearly shown that the aggregation of nano-Al was more serious than that of the micron-Al. The dispersion of nano-Al in RDX is also influenced by the aggregation of nano-Al. The micron-Al dispersed more evenly than the nano-Al.

### 3.2. Test and Analysis of Surface Energy and Adhesion Work of Materials

To further investigate the surface characteristics between nano-Al/micron-Al, RDX and paraffin wax, the four reagents are tested by the probe method. The obtained results are listed in Table 5.

In the two kinds of explosives, paraffin wax mainly plays the role of the desensitizer. Although the amount of paraffin wax is very small, the paraffin wax can be uniformly coated on the surface of explosive particles through appropriate preparation methods and induce the effect of reducing the mechanical sensitivity of the explosives [32]. The adhesion work between paraffin wax and other components can be further calculated using the surface energy of each component that is listed in Table 5. The adhesion work is defined as the reversible thermodynamic work required to separate the interface from the equilibrium state of two phases to infinity distance. When the two separated phases are identical, it is called work of cohesion. Adhesion work and cohesion work are the most powerful parameters to indicate the interaction between two phases.

According to this concept, the thermodynamic expressions of bonding work *W_AB_* and cohesion work *W_CA_* for A and B materials can be obtained:*W_AB_* = *γ_A_* + *γ_B_* − *γ_AB_*(1)
*W_AC_* = 2*γ_A_*(2)
where *γ_A_* and *γ_B_* are the surface free energies of the two materials, respectively, and *γ_AB_* is their interfacial free energy. In addition, according to the concepts of adhesion work and cohesion work, the reduction in Gibbs free energy per unit area when two separate phases approaching each other to form an interface is equal to the adhesion work and cohesion work.
Δ*G_AB_* = −*W_AB_*(3)
Δ*G_CA_* = −*W_CA_*(4)

Therefore, based on the thermodynamic point, it could be seen that the increase in adhesion work enlarges the interfacial tension between the two phases. The interfacial tension can be calculated by Formula (5) [32]. The calculated adhesion work of paraffin wax with RDX and different aluminum is provided in Table 6.
*γ_AB_* = [(*γ_A_^d^*)^1/2^ − (*γ_B_^d^*)^1/2^]^2^ + [(*γ_A_^p^*)^1/2^ − (*γ_B_^p^*)^1/2^]^2^(5)

According to Table 6, the adhesion work of paraffin wax with nano-Al is obviously higher than that of paraffin wax with micron-Al. The results show that paraffin wax is more easily coated on the surface of nano-Al than on the micron-Al. The results are also confirmed by the surface energy of Al in explosive 1 (nano-Al), which proves that the surface energy is lower than that in explosive 2 (micron-Al). Further investigation also shows that the adhesion work of paraffin wax to nano-Al is basically the same as that of paraffin wax-RDX. As the specific surface area of nano-Al is much larger than that of RDX, it can be inferred that in explosive 1 more paraffin wax is coated on the surface of nano-Al. Similarly, the correctness of the inference is also verified by the result that the weight and atomic percentage of the carbon in explosive 1 is higher than that in explosive 2.

### 3.3. Analysis of Adsorption Energy between Explosive Components

By calculating the adsorption energy of the main components of the explosives, the corresponding energy between the individual components and the combined components are obtained and shown in Table 7.

Based on the results in Table 7, the corresponding adsorption energy of paraffin wax to different Al powder and RDX can be obtained and the results are shown in Table 8.

It can be seen from the adsorption energy that, in the explosive system of RDX and Al powder, paraffin is more likely to be adsorbed on Al powder surface when the particle size of Al powder decreases from microns to nanometers accompanied with the specific surface area enlarged more than 10^4^ times. Therefore, in the explosive system containing nano-Al, the paraffin wax is mainly coated on the surface of the nano-Al powder [33]. Thus, in the explosive system containing micron-Al, the paraffin wax is mainly coated on the surface of the main explosive RDX.

### 3.4. Test and Analysis of Impact and Friction Sensitivity of Explosives

Based on the measurement of morphology, surface elements, surface energy of the explosives and the calculation of adhesion work, it could be seen that the component uniformity of explosive 1 is poor despite the components of explosive 1 and 2 being the same. The low mechanical sensitivity is the primary requirement for the employment of mixed explosives. Therefore, the mechanical sensitivity of explosive 1 and 2 are investigated and the results are shown in Table 9. As a comparison, the impact and friction sensitivity of RDX and RDX + paraffin wax were also provided.

Table 9 clearly shows that the impact sensitivity of RDX+ paraffin wax is very close to that of RDX, which means that the paraffin wax and RDX have good compatibility. The wax has almost no influence on the sensitivity of RDX [34]. Compared to the RDX + paraffin wax, the impact sensitivity of explosive 2 that is composed of micron-Al is the lowest of all the materials. This should be attributed to the micron-Al that could not be deformed when the explosive 2 was impacted and the deformation of micron-Al could not ignite RDX, which eventually decreased the impact sensitivity of explosive 2. On the other hand, the impact sensitivity of explosive 1 is much higher than that of explosive 2 and RDX accompanied with the friction sensitivity is also much higher than that of explosive 1. This should be attributed to the interface and microstructural characteristics of RDX-based explosives with nano-Al. As the specific surface area of nano-Al is much larger than that of RDX particles, more paraffin wax is coated on the surface of nano-Al, which means that the amount of RDX particles in explosive 1 are coated with less paraffin wax. To explosive 2, with the same amount of paraffin wax, micron-Al is more uniformly attached on the surface of RDX particles. Under the impact of the drop hammer, RDX is lubricated and buffered by micron-Al, which results in the different impact sensitivity of the two explosives. To friction sensitivity, explosive 1 is higher than explosive 2, which should be caused by nano-Al. The nano-Al has higher specific surface area than micron-Al, which is easier to be ignited by friction. Compared with RDX and RDX + paraffin wax, the friction sensitivity of explosive 1 is higher than that of RDX + paraffin wax and RDX, which should be caused by the adsorption of paraffin wax between nano-Al and RDX.

## 4. Conclusions

Based on the investigations of microstructures, the surface–interface coating effect and calculations of material molecular models of RDX-based explosives containing nano-sized and micron-sized aluminum powder, it is found that the agglomeration tends to occur in nano-Al with a larger specific surface area. At the same time, the binder of the explosive formulation (especially the paraffin-based binder) can be more easily coated on the surface of nano-Al powder, which reduces the impact sensitivity of the explosive. The results of the paper provide a safer way to design high-performance explosives containing nano-Al powder. The paraffin wax should have the performance of easily adsorbing on RDX particles and forming a good coating on RDX surface. Newly designed preparation methods should consider how to coat the paraffin on the surface of the energetic material particles firstly, and then uniformly mix them with nano-Al.

## Figures and Tables

**Figure 1 nanomaterials-11-02182-f001:**
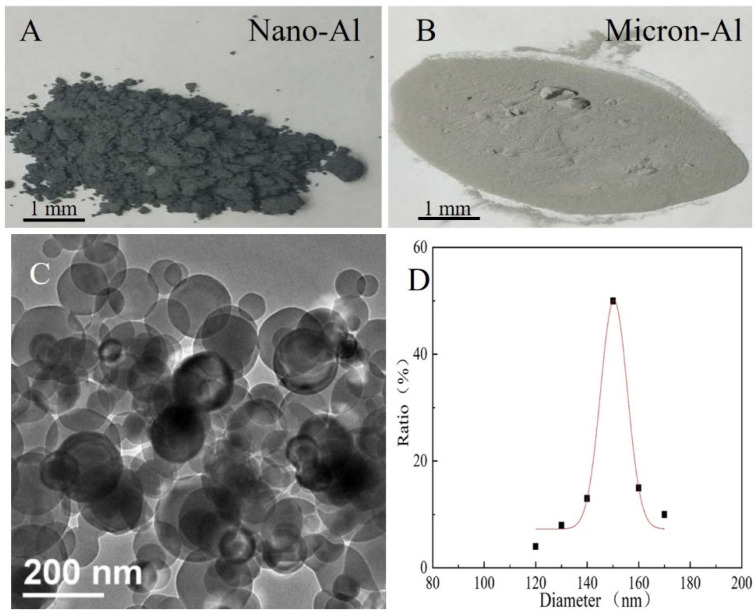
Photography of nano-Al (**A**) and micron-Al (**B**) powders. (**C**) TEM tests of nano-Al and (**D**) the distribution of the diameter of nano-Al powder.

**Figure 2 nanomaterials-11-02182-f002:**
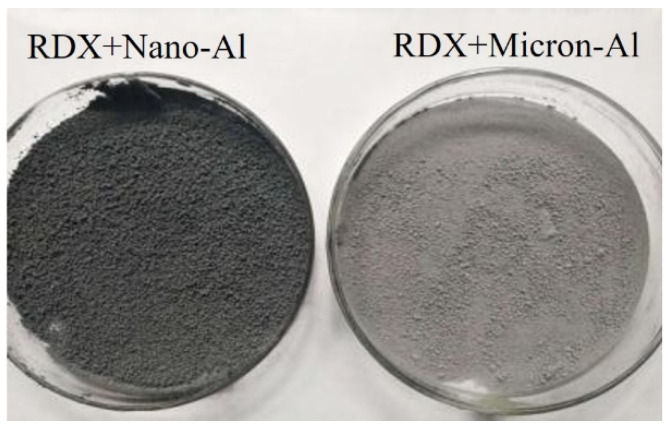
Photography of RDX with Nano-Al and Micron-Al.

**Figure 3 nanomaterials-11-02182-f003:**
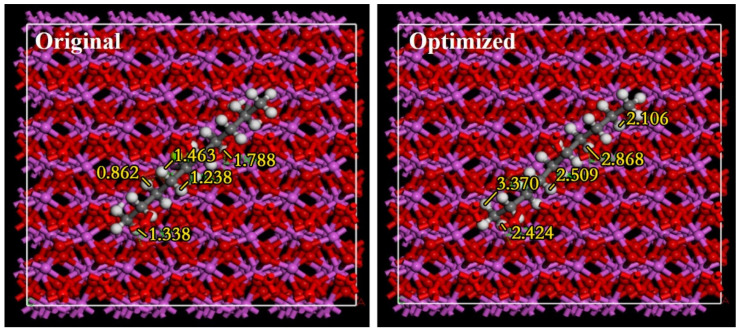
Original and optimized molecular structure of paraffin adsorbed on Al_2_O_3._

**Figure 4 nanomaterials-11-02182-f004:**
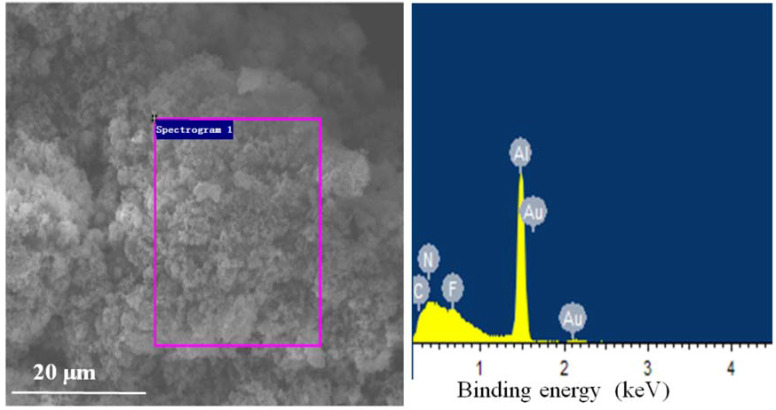
SEM image and EDS spectra of explosive 1.

**Figure 5 nanomaterials-11-02182-f005:**
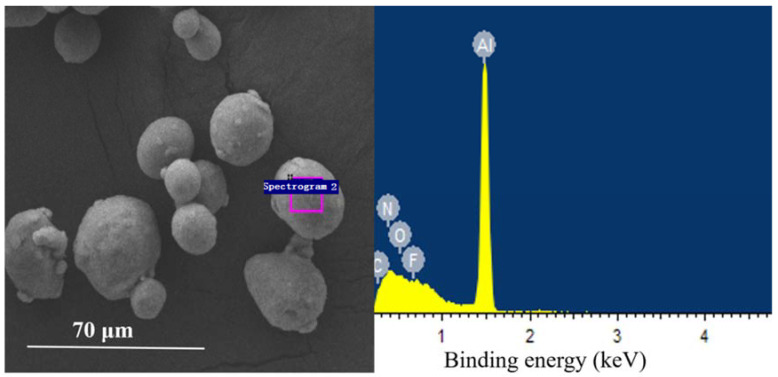
SEM image and EDS spectra of explosive 2.

**Figure 6 nanomaterials-11-02182-f006:**
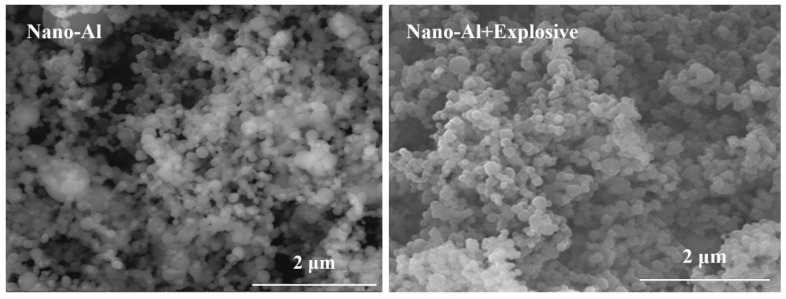
SEM images of nano-Al and explosive containing nano-Al.

**Figure 7 nanomaterials-11-02182-f007:**
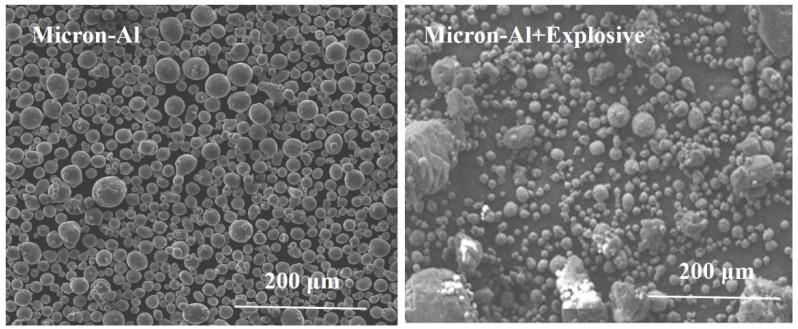
SEM images of micron-Al and explosive containing micron-Al.

**Table 1 nanomaterials-11-02182-t001:** Components and proportions of the two kinds of explosives samples.

Name	Composition and Weight Ratio (%)
Explosive 1	nano-Al 30
	RDX 67
	Paraffin wax 3
Explosive 2	micron Al 30
	RDX 67
	Paraffin wax 3

**Table 2 nanomaterials-11-02182-t002:** Energy calculation of different Al_2_O_3_ sections.

Section No.	Crystal Plane						
	(110)	(101)	(011)	(111)	(100)	(010)	(001)
1	−283.38324 eV	−293.38312 eV	−283.38323 eV	−293.0993 eV	−289.0031863 eV	−286.31209 eV	−286.3075026 eV
2	−278.86804 eV	−278.887853 eV	−278.886804 eV	−283.32663 eV	−289.0031755 eV	-	−286.3075033 eV
3	−286.58172 eV	−286.58182 eV	−286.58891 eV	−283.65849 eV	−289.0031783 eV	−286.0970793 eV	−286.0969497 eV
4	−286.65099 eV	−286.65109 eV	−286.65488 eV	−293.09043 eV	−289.0031823 eV	−289.0148331 eV	−289.0031925 eV
5	−278.31891 eV	−278.31902 eV	−278.31895 eV	−283.31319 eV	−289.0031799 eV	−289.4154393 eV	−289.0031915 eV
6	−293.38288 eV	−283.38337 eV	−293.39229 eV	−293.0993 eV	−289.0031804 eV	−286.3075868 eV	−286.3074286 eV
7	-	-	-	−293.0993 eV	−289.0031854 eV	−292.1069331 eV	−292.09674 eV

**Table 3 nanomaterials-11-02182-t003:** Elemental weight and atomic percentage of explosive 1.

Element	Weight Percentage	Atomic Percentage
C	38.10	54.53
N	7.38	9.06
F	6.24	5.64
Al	48.35	30.76
Total	100.00	100

**Table 4 nanomaterials-11-02182-t004:** Elemental weight and atomic percentage of explosive 2.

Element	Weight Percentage	Atomic Percentage
C	28.55	44.89
N	7.38	9.96
O	−0.41	−0.48
F	1.68	1.67
Al	62.80	43.97
Total	100.00	100

**Table 5 nanomaterials-11-02182-t005:** Surface energy of each explosive component.

Reagent	Surface Energy (mJ)	Polar Component (mN/m)	Dispersion Component (mN/m)	Probe Liquid
RDX	41.29	34.44	6.85	water and formamide
Nano-Al	40.01	12.52	27.49	glycol and glycerol
Micron-Al	18.55	5.48	13.07	glycol and glycerol
Paraffin wax	14.82	12.78	2.04	water and glycol

**Table 6 nanomaterials-11-02182-t006:** Adhesion work between paraffin wax with RDX and aluminum with different size.

Materials	Adhesion Work (mJ)
Paraffin with RDX	49.40
Paraffin with nano-Al	40.31
Paraffin with micron-Al	27.06

**Table 7 nanomaterials-11-02182-t007:** Parameter setting and the corresponding calculated energy of each component.

Structure	Van der Waals Force Correction Method	K-Point Grid	Calculation Accuracy	Adsorption Energy (eV)
Al_2_O_3_	DFT-D3	9 × 9 × 1	Default precision	−298.49886
paraffin	DFT-D3	1 × 1 × 1	Default precision	−173.074
RDX	DFT-D3	3 × 3 × 1	Default precision	−4173.2264
Al_2_O_3_ + paraffin	DFT-D3	1 × 1 × 1	Default precision	−4950.088
RDX + paraffin	DFT-D3	2 × 1 × 1	Default precision	−4328.7048

**Table 8 nanomaterials-11-02182-t008:** Corresponding adsorption energy between components.

Structure	Adsorption Energy (eV)
Paraffin-Micro-Al	−0.9301
Paraffin-Nano-Al	−2.312
Paraffin-RDX	17.5956

**Table 9 nanomaterials-11-02182-t009:** Impact and friction sensitivity of the two explosive samples.

+	Impact Sensitivity (cm)	Friction Sensitivity (%)
Explosive 1	27.5	8
Explosive 2	97.7	0
RDX + paraffin wax	94.5	0
RDX	95.7	84~76

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
