# Peer review of "Effects of Nano Aluminum Powder on the Mechanical Sensitivity of RDX-Based Explosives"

_nanomaterials, 2021, doi:10.3390/nano11092182_

Round 1
Reviewer 1 Report
Submitted article “Effects of Nano Aluminum Powder on the Mechanical Sensitivity of RDX-based Explosives” by Jun Dong , Weili Wang , Xiaofeng Wang , Qiang Zhou , Run Miao , Maohua Du , Bo Tan , Yuanjing Wang ,Tengyue Zhang, Yafei Li, Fangjie Cao fits the profile of the journal and can be printed with minor revisions.
Reviewer's note.
Table 9 is key in this study. I believe that the article can be improved if the results of studying the impact and friction sensitivity of a sample of used RDX and a sample of a mixture of RDX and paraffin will be added to Table 9. Then the authors of the article need to discuss jointly the new results with the explosive 1 and explosive 2 sensitivity results discussed in the article.
Author Response
Authors’ response:
Based on the comments, it could be seen that the reviewer is an expert in energetic materials and also familiar with the processing of papers. The impact and friction sensitivity of RDX and RDX + paraffin wax are provided in Table 9. And the influences are also illustrated in Page 10 and 11.
Page 10:
Table 9. Impact and friction sensitivity of the two explosive samples
|
Materials |
Impact sensitivity (cm) |
Friction sensitivity (%) |
|
Explosive 1 |
27.5 |
8 |
|
Explosive 2 |
97.7 |
0 |
|
RDX+ paraffin wax |
42.5 |
0 |
|
RDX |
95.7 |
84~76 |
Table 9 clearly shows that the impact sensitivity of RDX is much higher than that of RDX + paraffin wax, which means that the paraffin wax greatly reduced the sensitivity. The decrease of impact sensitivity of RDX should be attributed to the paraffin wax that is easily adherent to the crystal surface RDX and eventually inhibit the effect between RDX molecules [34]. But compared to the RDX + paraffin wax, the impact sensitivity of explosive 2 that composed by micron-Al is the highest of all the materials. This should be attributed the micron-Al that could be deformed when the explosive 2 was impacted and the deformation of micron-Al could ignite RDX, which eventually enhanced the impact-sensitivity of explosive 2.
Page 11:
To friction sensitivity, the explosive 1 is higher than explosive 2, which should be caused by nano-Al. The nano-Al has higher specific surface area than micron-Al, which is easier to be ignited by friction. Compared with RDX and RDX + paraffin wax, the friction sensitivity of explosive 1 is lower than that of RDX + paraffin wax but higher than that of RDX, which should be caused by the adsorption of paraffin wax between nano-Al and RDX.
Reviewer 2 Report
The authors reported the effect of aluminium powder on the friction and impact sensitivities of the aluminium/wax/RDX energetic mixture. Both nanometric and micronic aluminium powders were tested to determine a possible size effect. The investigation is mainly supported by a mathematical approach (software). The study is interesting and shows new results that could be used by pyrotechnicians in the design of energetic mixture formulations with various mechanical sensitivities. The large number of references written in Chinese (30%) makes it difficult to appreciate this paper. A more relevant journal for energetic materials would be more appropriate to disseminate this study (Journal of Hazardous Materials, Propellants Explosives, Pyrotechnics, and Journal of Energetic Materials). It is
therefore the editor’s right of discretion whether to consider publish this paper or not.
Author Response
It could be seen that the reviewer has carefully examined our paper and give the constructive comments. According to the comments we added 3 pieces of new references 32, 33 and 34 in the paper. The references are also listed in Page 13.
Page 13:
[32] Chen Binbin, Xia Zhixun, Huang Liya, Ma Likun. Characteristics of the combustion chamber of a boron-based solid propellant ducted rocket with a chin-type inlet. Aerospace Science and Technology 2018, 82-83, 210-219.
[33] Li Chao, Hu Chunbo, Deng Zhe, Hu Xu, Li Yue, Wei Jinjia. Dynamic ignition and combustion characteristics of agglomerated boron-magnesium particles in hot gas flow. Aerospace Science and Technology, 2012, 110, 106478.
[34] Yang Qi, Liu Xiangyu, Qu Xiaoni, Wei Qing, Xie Gang, Chen Sanping, Gao Shengli. High-energy metal–organic frameworks (HE-MOFs): Synthesis, structure and energetic performance, Coordination Chemistry Reviews, 2016, 307, 292-312
Round 2
Reviewer 2 Report
In this work, the authors studied the effect of adding aluminium powder on the mechanical sensitivity of an RDX-based energetic mixture. Two aluminium particle sizes (nano and micron) were tested. The paper does not seem suitable for publication in Nanomaterials. A journal more dedicated to pyrotechnics would be more appropriate (Journal of Hazardous Materials, Propellants Explosives, Pyrotechnics, and Journal of Energetic Materials) and in line with all the references cited in this article. In addition, the discussion section of the article requires additional work to warrant its publication. The following specific comments may help authors to "improve" their manuscript.
1/ A scale bar should be added to the photographs of aluminium powders. In addition, an HRTEM analysis is need to observe the core/shell (alumina/aluminium) structure of the powders. A particle size distribution of the nano- and micron-sized aluminium powders would be appropriate to observe the agglomeration state of the two powders. The specific surface area values of the two Al powders and the RDX explosive should be given in the manuscript together with a description of the procedure.
2/ FTIR spectra are needed to analyse the ternary mixture and the type of bonds existing in the ternary mixture. Is the RDX explosive soluble in petroleum ether?
3/ What does "semi-dry state" mean in section 2.2 Preparation of the explosive? How do the authors control this parameter?
4/ On page 4, section 2.3, could the two paragraphs on mechanical testing be more detailed? How do these devices work?
5/ Same remark concerning paragraph 2.4.2. I understand why Al2O3 is taken into account in the calculations (and not Al) but if the authors do not characterise the aluminium powders using the TEM technique, it could be difficult for the readers to understand.
6/ The legend of table 2 is written in bold.7/ In Figure 3, it is difficult to see and read the values in green.
8/ The term "well recognised" (page 1, Introduction section), "dosage" (page 4, performance test) are not appropriate.
9/ Page 2, line 60, "powder" instead of "power".
10/ Figures 4 and 5, "SEM image and EDS spectra" instead of "SEM and EDS". A similar scale bar is required to compare microstructures.
11/Page 7, line 217, "images" instead of "test results".
12/ Table 3, Au (gold) can be excluded as it is not a component of the various as-prepared explosives? The gold is probably from the coating applied to the explosive particles for SEM analysis?
13/ For the two images in figure 6, a similar scale would be preferable to observe the differences between the nano-Al and the Nano-Al+Explosive. Same remark for figures 7.
14/ Paragraph 3.2 (pages 8-9) needs some references. Equation (5) should be a justified reference. What do 1 and 2 mean in this equation?
15/ In paragraph 3.3, how is the absorption energy calculated (table 8)? In this table, is Al for nano or micron powder? The paragraph following this table needs to be improved. Why do the authors say that wax is more easily absorbed on Al nanoparticles than on Al microparticles?
16/ Paragraph 3.4 needs to be explained. For example, reading the value of the impact sensitivity of explosive 1 (which is lower than that of explosive 2), one may believe that it is more sensitive since the explosion reaction takes place with a low hammer drop (compared to explosive 2). Can the authors explain this?
Author Response
Reviewer
In this work, the authors studied the effect of adding aluminium powder on the mechanical sensitivity of an RDX-based energetic mixture. Two aluminium particle sizes (nano and micron) were tested. The paper does not seem suitable for publication in Nanomaterials. A journal more dedicated to pyrotechnics would be more appropriate (Journal of Hazardous Materials, Propellants Explosives, Pyrotechnics, and Journal of Energetic Materials) and in line with all the references cited in this article. In addition, the discussion section of the article requires additional work to warrant its publication. The following specific comments may help authors to "improve" their manuscript.
Reviewer:1/ A scale bar should be added to the photographs of aluminium powders. In addition, an HRTEM analysis is need to observe the core/shell (alumina/aluminium) structure of the powders. A particle size distribution of the nano- and micron-sized aluminium powders would be appropriate to observe the agglomeration state of the two powders. The specific surface area values of the two Al powders and the RDX explosive should be given in the manuscript together with a description of the procedure.
Author response: Thanks for the reviewer’s constructive comments. It could be seen that the reviewer is an expert in the field of energetic materials. We have revised the paper according to reviewer’s comments. The TEM tests of nano-Al are provided in Figure 1. The specific surface area test methods and the results are supplied in page 3 paragraph 2.
Page 3 paragraph 3 and Figure 1:
Figure 1C showed the TEM test results of nano-Al powder, which clearly exhibited that the nano-Al were in ball-like structure. The diameter of the nano-Al are also examined and the results are showed in Figure 1D. It is obviously showed that the diameter of nano-Al powder located in a large range despite the major diameter located in about 150 nm.
Figure 1. Photography of Nano-Al (A) and Micron-Al (B) powders. (C) TEM tests of nano-Al and (D) the distribution of the diameter of nano-Al powder.
Page 3 paragraph 2
N2 adsorption–desorption measurements were carried out at 77 K using a Quantachrome Autosorb gas-sorption system to determine the specific surface area of nano-Al, micro-Al and RDX, separately. The specific surface area of nano-Al, micron-Al and RDX are 22.22, 0.55 and 0.344 m2/g. The transmission electron microscopy (TEM) tests of nano-Al was operated by using a Hitachi (H 9000 NAR).
Reviewer: 2/ FTIR spectra are needed to analyse the ternary mixture and the type of bonds existing in the ternary mixture. Is the RDX explosive soluble in petroleum ether?
Author response: Thanks for the reviewer’s constructive comments. It could be seen the reviewer is very familiar with the tests and analysis of materials test methodology. But, at present, we cannot test the FTIR. We will provide the related FTIR tests in the following work.
Based on our tests, we don’t observed the soluble of RDX in petroleum ether.
Reviewer:3/ What does "semi-dry state" mean in section 2.2 Preparation of the explosive? How do the authors control this parameter?
Author response: Thanks for the reviewer’s careful examination on the paper. The state of “semi-dry” is illustrate in the paper. To be honest, the state of “semi-dry” control is based on the experience. It still can be briefly described though it is not very strict.
Page 4 paragraph 1: Then the mixture was poured out and the solvent was volatilized until the explosive reached the semi-dry state, which means that there is no petroleum ether could be volatized anymore but the state still looks a little wet.
Reviewer:4/ On page 4, section 2.3, could the two paragraphs on mechanical testing be more detailed? How do these devices work?
Author response: Thanks for the reviewer’s careful examination. The test detail of the impact sensitivity tests were provided.
Page 4, section 2.3: The impact sensitivity of explosives is measured by the calibrated H3.5-10W drop-hammer impact sensitivity instrument. The weight of the drop hammer was 10 kg and the weight of explosive is 50 mg. The examination was conducted according to the drop height of each test. The 50% explore drop height of the tests is believed as the impact sensitivity. The tests are divided into two groups and each group has 25 rounds. And the final result is the even value of all the test rounds in each group.
Reviewer:5/ Same remark concerning paragraph 2.4.2. I understand why Al2O3 is taken into account in the calculations (and not Al) but if the authors do not characterise the aluminium powders using the TEM technique, it could be difficult for the readers to understand.
Author response: Thanks for the reviewer’s careful examination. The Al2O3 is considered as the surface of Al-powder due to the oxidation of Al in atmosphere. On the other hand, the calculation also should set an end for the software. The oxide layer of Al2O3 is only a few atomic layer, which make it difficult to be obtained by TEM tests. We will make more effect in the following work to detect the oxide layer of Al2O3.
Reviewer:6/ The legend of table 2 is written in bold.
Author response: Thanks for the reviewer’s careful examination. We have revised it.
Reviewer:7/ In Figure 3, it is difficult to see and read the values in green.
Author response: Thanks for the reviewer’s careful examination. We have re-draw the picture and the data were reformed.
Figure 3. Original and optimized molecular structure of paraffin adsorbed on Al2O3
Reviewer:8/ The term "well recognised" (page 1, Introduction section), "dosage" (page 4, performance test) are not appropriate.
Author response: Thanks for the reviewer’s careful examination. We have revised them according to the comments.
Page 2:The well recognized is deleted.
Page 4:The “dosage” has been changed to “weight”.
Reviewer:9/ Page 2, line 60, "powder" instead of "power".
Author response: Thanks for the reviewer’s careful examination. The “power” has been revised to “powder”.
Reviewer:10/ Figures 4 and 5, "SEM image and EDS spectra" instead of "SEM and EDS". A similar scale bar is required to compare microstructures.
Author response: Thanks for the reviewer’s careful examination. The title of Figure 4 and5 has been revised.
Page 7: Figure 4. SEM image and EDS spectra of explosive 1
Figure 5. SEM image and EDS spectra of explosive 2
Reviewer:11/Page 7, line 217, "images" instead of "test results".
Author response: Thanks for the reviewer’s careful examination. The “test results” has been revised to “image”.
Reviewer:12/ Table 3, Au (gold) can be excluded as it is not a component of the various as-prepared explosives? The gold is probably from the coating applied to the explosive particles for SEM analysis?
Author response: Thanks for the reviewer’s careful examination. The Au is originate from the SEM tests according to the reviewer’s comments we have removed the element Au from Table 3. The corresponding content of each element was also recalculated.
Table 3. Elemental weight and atomic percentage of explosive 1
|
Element |
Weight percentage |
Atomic percentage |
|
C |
38.10 |
54.53 |
|
N |
7.38 |
9.06 |
|
F |
6.24 |
5.64 |
|
Al |
48.35 |
30.76 |
|
Total |
100.00 |
100 |
Reviewer:13/ For the two images in figure 6, a similar scale would be preferable to observe the differences between the nano-Al and the Nano-Al+Explosive. Same remark for figures 7.
Author response: Thanks for the reviewer’s careful examination. The remark in both Figure 6 and 7 have been revised.
Figure 6. SEM images of nano-Al and explosive containing nano-Al.
Figure 7. SEM images of micron-Al and explosive containing micron-Al
Reviewer:14/ Paragraph 3.2 (pages 8-9) needs some references. Equation (5) should be a justified reference. What do 1 and 2 mean in this equation?
Author response: Thanks for the reviewer’s careful examination. We have supplied the reference and the revised the equation.
[33] D. W. VAnkrevelen, K. te Nijenhuis. Proparties of Polymers. Their Correlation with Chemical Structure. Their Numerical Estimation and Prediction from Additive Group Contributions. [M] ELSEVIER, 1981.
Page 9 γAB=[(γAd)1/2-(γBd)1/2]2+[(γAp)1/2-(γBp)1/2]2 (5)
Reviewer:15/ In paragraph 3.3, how is the absorption energy calculated (table 8)? In this table, is Al for nano or micron powder? The paragraph following this table needs to be improved. Why do the authors say that wax is more easily absorbed on Al nanoparticles than on Al microparticles?
Author response: Thanks for the reviewer’s careful examination. The absorption energy of nano-Al and micron-Al to paraffin wax were calculated separately. Based on the calculated absorption energy, we deduced that wax is more easily absorbed on Al nano-particles.
Page 10 Table 8
Table 8. Corresponding adsorption energy between components
|
Structure |
Adsorption energy (eV) |
|
Paraffin – Micro-Al |
-0.9301 |
|
Paraffin – Nano-Al |
-2.312 |
|
Paraffin - RDX |
17.5956 |
Reviewer:16/ Paragraph 3.4 needs to be explained. For example, reading the value of the impact sensitivity of explosive 1 (which is lower than that of explosive 2), one may believe that it is more sensitive since the explosion reaction takes place with a low hammer drop (compared to explosive 2). Can the authors explain this?
Author response: Thanks for the reviewer’s careful examination. We have made a mistake in the last version. Now, we have corrected the mistake and explained the phenomenon agian.
Page 11 paragraph 1: Table 9 clearly shows that the impact sensitivity of RDX+ paraffin wax is very close to that of RDX, which means that the paraffin wax and RDX have good compatibility. The wax almost make no influence on the sensitivity of RDX [35]. Compared to the RDX + paraffin wax, the impact sensitivity of explosive 2 that composed by micron-Al is the lowest of all the materials. This should be attributed the micron-Al that couldnot be deformed when the explosive 2 was impacted and the deformation of micron-Al couldnot ignite RDX, which eventually decreased the impact-sensitivity of explosive 2. On the other hand, the impact sensitivity of explosive 1 is much higher than that of explosive 2 and RDX accompanied with the friction sensitivity is also much higher than that of explosive 1.
